## PERSPECTIVE

# Can aerobic exercise really be a 'warm-up' for brown adipose tissue?

Milena Schönke[1,2,3]
and Brendan M. Gabriel[3,4]

[1]*Department of Medicine, Division of Endocrinology, Leiden University Medical Center, Leiden, The Netherlands*
[2]*Einthoven Laboratory for Experimental Vascular Medicine, Leiden University Medical Center, Leiden, The Netherlands*
[3]*Department of Physiology and Pharmacology, Integrative Physiology, Karolinska Institutet, Stockholm, Sweden*
[4]*Aberdeen Cardiovascular & Diabetes Centre, The Rowett Institute, University of Aberdeen, Aberdeen, United Kingdom*

Email:     Brendan.gabriel1@abdn.ac.uk, Brendan.gabriel@ki.se

Edited by: Michael Hogan & Bettina Mittendorfer

Linked articles: This Perspective article highlights an article by Kim *et al.* To read this paper, visit https://doi.org/10.1113/JP282999.

The peer review history is available in the Supporting information section of this article (https://doi.org/10.1113/JP283087#support-information-section).

Obesity is a condition with a high and growing prevalence worldwide. Performing regular exercise has many health benefits and has long been thought of as a key adjunct to weight-loss interventions such as diet and pharmaceutical treatment. However, it should be noted that exercise has a modest effect on weight loss when performed as a stand-alone intervention. On the other hand, it is possible that exercise may have a more robust effect on preventing weight gain than inducing weight loss, particularly as a stand-alone intervention.

Exercise affects many tissues and metabolic pathways throughout the body; however, it is unclear exactly how it is able to act as a beneficial adjunct to weight-loss interventions, and play a role in preventing weight gain. One putative, contributory pathway may be exercise-induced activation of thermogenesis in brown adipose tissue (BAT). BAT has been identified as a potential therapeutic target for weight-loss strategies for several decades. This tissue uses redundant cycling of an electrochemical gradient, termed 'uncoupling' to produce heat (thermogenesis). BAT contains a relatively high number of mitochondria expressing the thermogenic protein 'mitochondrial brown fat uncoupling protein 1' (UCP1), which is key in this process. Safe and robust activation of this tissue would clearly be a promising strategy for weight loss due to its metabolism of redundant chemical energy resulting in a relatively benign and manageable side-effect of increased thermogenesis. However, much foundational work in the BAT field was performed in rodents, and questions remain over the translatability of results from these rodent studies to human healthcare. These translatability issues include the fact that humans have far less relative BAT, and a reduced reliance on BAT as a source of thermogenesis compared with mice. Additionally, the housing temperature of rodents in laboratory studies may confound the interpretation of many studies of BAT activation. There has also been much debate as to whether exercise robustly induces thermogenesis in BAT. This is another area where translatability from rodents to humans may not be straightforward. Recent data from human studies suggest that exercise does not inherently activate thermogenesis in BAT (Motiani et al., 2019 and unpublished data from a randomized-control trial (Sanchez-Delgado et al., 2015)). However, many human exercise studies are performed indoors and it is possible there may be a synergistic effect of outdoor exposure (i.e. sunlight and cold exposure) and exercise (Gabriel & Zierath, 2021), although this has yet to be determined.

Kim, Kim and Seong sought to further elucidate this area in their recent study (Kim et al., 2022). The current study assessed whether voluntary wheel-running could change *Ucp1* expression in BAT. These exercising mice saw increased *Ucp1* mRNA expression in white adipose tissue and BAT. This is indicative of increased thermogenesis in BAT, although this was not directly measured. The authors then treated *in vitro* BAT cells with harvested serum from the exercised mice. *Ucp1* expression was increased in these BAT cells, indicating that a factor in serum was responsible for the increased *Ucp1* expression in the exercising mice's BAT tissue (Fig. 1). The authors hypothesized that AMPK activation in skeletal muscle could be essential to secretion of this serum-signalling molecule. Thus, they used pharmacological compounds to activate AMPK *in vitro* in skeletal muscle cells and collected the media from these cells. Again, *Ucp1* expression increased in BAT cells treated with this conditioned media. The authors then performed mechanistic experiments using siRNA to silence the two isoforms of the catalytic subunit of AMPK, thereby supporting the specificity of AMPK's role in this pathway (Fig. 1). It has been previously shown that AMPK activation in BAT itself is important in the regulation of thermogenesis (van Dam et al., 2015), thus raising the possibility of a signalling molecule with roles in both BAT and skeletal muscle. The authors went on to speculate that IL-6 secretion from skeletal muscle might play a role in the upregulation of *Ucp1* in BAT. However, their evidence suggests factors other than IL-6 are more important in this phenomenon since there was no ablation of *Ucp1* activation in BAT cells after treatment with IL-6 neutralizing antibody-supplemented conditioned media.

Amongst the most interesting data in this study is the effect of cultured media from skeletal muscle cells upon activation of *Ucp1* expression in BAT. It would be very impactful to identify the factor or factors which engender this effect. Future studies should aim to use metabolomics or proteomics analysis to further investigate AICAR- or exercise-induced changes in skeletal muscle media or serum. Ideally, human data supporting this effect would aid the translatability of these findings.

In summary, questions remain in this field about the physiological and clinical translatability of BAT activation from rodent studies to humans. However, given the multifactorial, beneficial roles that

The Journal of Physiology

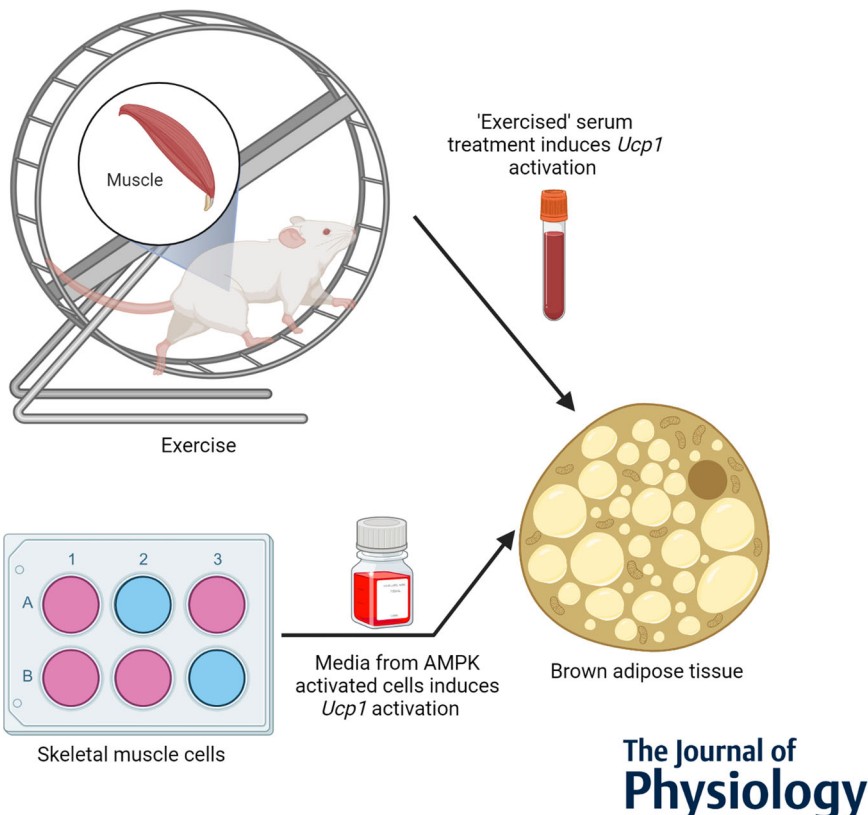

**Figure 1. A schematic of the methods and findings of Hye Jin Kim, Youn Ju Kim, 2022. Created in Biorender**

exercise has in the body, further elucidating the factors that allow cross-talk between exercised muscle and adipose tissue would be extremely valuable to our efforts to treat metabolic disease and obesity.

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

## Additional information

### Competing interests

None.

### Author contributions

Both authors have read and approved the final version of this manuscript and agree to be accountable for all aspects of the work in ensuring that questions related to the accuracy or integrity of any part of the work are appropriately investigated and resolved. All persons designated as authors qualify for authorship, and all those who qualify for authorship are listed.

### Funding

B.M.G. was supported by a fellowship from the Novo Nordisk Foundation (NNF19OC0055072). M.S. was supported by the Novo Nordisk Foundation (NNF18OC0032394).

### Keywords

brown adipose, exercise physiology, thermogenesis

## Supporting information

Additional supporting information can be found online in the Supporting Information section at the end of the HTML view of the article. Supporting information files available:

**Peer Review History**

