## [Peer Review History · The Journal of Physiology]

Can aerobic exercise really be a 'warm-up' for brown adipose tissue?

Brendan Gabriel and Milena Schönke
DOI: 10.1113/JP283087

Corresponding author(s): Brendan Gabriel (brendan.gabriel1@abdn.ac.uk)

The following individual(s) involved in review of this submission have agreed to reveal their identity: Je Kyung Seong (Referee #1)

Review Timeline:

Submission Date:

01-Apr-2022

Accepted:

08-Apr-2022

Senior Editor: Michael Hogan

Reviewing Editor: Bettina Mittendorfer

Transaction Report:

Dear Dr Gabriel,

Re: JP-P-2022-283087 "Can aerobic exercise really be a 'warm-up' for brown adipose tissue?" by Brendan Gabriel Milena Schönke

I am pleased to tell you that your invited Perspective article has been accepted for publication in The Journal of Physiology.

IMPORTANT

We only have a PDF of your article. Our typesetter will require a Word file (of all text) and a separate high resolution figure file. Please can you send these to the office as soon as possible at: jp@physoc.org

Thank you.

NEW POLICY: In order to improve the transparency of its peer review process The Journal of Physiology publishes online as supporting information the peer review history of all articles accepted for publication. Readers will have access to decision letters, including all Editors' comments and referee reports, for each version of the manuscript and any author responses to peer review comments. Referees can decide whether or not they wish to be named on the peer review history document.

The last Word version of the paper submitted will be used by the Production Editors to prepare your proof. When this is ready you will receive an email containing a link to Wiley's Online Proofing System. The proof should be checked and corrected as quickly as possible.

All queries at proof stage should be sent to tjp@wiley.com

Thank you very much for your contribution to The Journal of Physiology.

Yours sincerely,

Michael C. Hogan
Senior Editor
The Journal of Physiology
<https://jp.msubmit.net>
<http://jp.physoc.org>
The Physiological Society
Hodgkin Huxley House
30 Farringdon Lane
London, EC1R 3AW
UK
<http://www.physoc.org>
<http://journals.physoc.org>

Reviewing Editor Comments:

Nice work
Thank you

Reviewer Comments:

Thank you for the well-discussed perspectives article on our manuscript. The findings of the our paper are well-summarized in this review, which emphasizes the need for more research into the implications of exercise-dependent brown fat. Future experiments that would certainly extend the findings are proposed (e.g. metabolomics or proteomics). This perspective aligns well with our manuscript.